# Multi-Disaster Integrated Risk Assessment in City Range—A Case Study of Jinan, China

**DOI:** 10.3390/ijerph20043483

**Published:** 2023-02-16

**Authors:** Jun Chang, Zuotang Yin, Zhendong Zhang, Xiaotong Xu, Min Zhao

**Affiliations:** 1College of Geography and Environment, Shandong Normal University, Jinan 250358, China; 2Jinan Survey and Mapping Institute, Jinan 250101, China

**Keywords:** multi-disaster integrated risk assessment, territorial spatial planning, Jinan city

## Abstract

Urban multi-disaster integrated risk assessment is an important part of urban sustainable development and territorial spatial planning. Based on the results of integrated risk assessment, the scientific and effective performance of disaster prevention and reduction can be effectively improved. This study determines a multi-disaster integrated risk assessment system. The system evaluates the hazard level of disasters, the exposure level of disaster bearing bodies, the vulnerability level of disaster bearing bodies, and the urban resilience level, and determines the city’s integrated risk level on this basis. Taking Jinan as an example, the risk, exposure, vulnerability, resilience, and integrated risk level of Jinan City were analyzed. The results show that the system reasonably analyzes the multi-disaster integrated risk level, and according to the assessment results, countermeasures for disaster prevention and suggestions for territorial spatial planning were put forward.

## 1. Introduction

The city is the center of regional social and economic development. In most countries, 70–80 percent of gross domestic product (GDP) is concentrated in cities [1]. Moreover, cities have a more dense population, clogged traffic, and vulnerable socio-economic than rural areas, which makes natural disasters more destructive in cities [2,3]. This feature is more pronounced in many developing countries, especially in Asia [2,3]. In developing countries, with the migration of rural populations to cities, challenges and opportunities coexist, which increases urban economies of scale but also brings greater pressure to cities [3]. The policies of territorial spatial planning and optimization in developing countries are relatively backward, the mechanism of disaster emergency response is not perfect, and sudden disasters are more likely to cause severe harm to local people and economies.

As the largest developing country in the world, China has more than 900 million people living in cities [4]. The high concentration of population limits the suitability of urban and territorial spatial development and causes a higher level of urban exposure and vulnerability to natural disasters. Danan GU [5] of the Population Division, United Nations Department of Economic and Social Affairs conducted a study on natural disasters in cities with populations of more than 300,000 in 2018. The study analyzed the risks of flood, drought, earthquake, geology, hurricane, and volcanic eruptions in 1860 cities around the world, as well as the vulnerability of economic losses caused by these six disasters. Among Chinese cities with a population of more than 300,000, the study found that many cities had exposure deciles of 8–10th and were highly exposed to floods, hurricanes, and cyclones, and the mortality vulnerability deciles and economic vulnerability deciles caused by drought, and flood were higher. This showed that natural disasters had a serious impact on the security of urban residents and economic development. Therefore, it is necessary to carry out a more in-depth assessment of the suitability of urban land and space development and to assess the disaster risk faced by cities from a more comprehensive perspective.

In previous evaluations, for example, in the study of flood risk assessment in the Day River Flood Diversion Area [6] and Jing River flood diversion district [7], the risk was defined as the product of hazard and vulnerability [6,8], that is, Risk = Hazard × Vulnerability, where the hazard is a function of the environmental damage or economic loss caused by natural or man-made disasters in a given area [7], vulnerability refers to the degree of loss or damage suffered by the population, buildings, economies, infrastructure, and other disaster bearers due to their own value content [9]. This approach combines risk with the natural (hazard) and social (vulnerability) characteristics of disasters, and to a certain extent has been recognized by scholars all over the world. However, the current disaster risk research has not formed a unified evaluation system [10], the above formula does not take into account the disaster exposure of the disaster-bearing body and the resilience of cities to disasters, there are still some deficiencies. Some scholars and organizations believe that disasters are “serious disruption of the functioning of a community or a society at any scale due to hazardous events interacting with conditions of exposure, vulnerability, and capacity, leading to one or more of the following: human, material, economic and environmental losses and impacts” [11]. The object of multi-disaster integrated risk assessment is a complex system composed of disaster risk, geographical environment (the disaster-bearing body), and exposure unit, and the recovery ability of different geographical environments is different [12,13,14]. Therefore, based on the concept of the resilient city [15,16,17], this study, considering disaster risk and disaster-bearing body vulnerability, assess the exposure of geographical factors to disasters and their resilience [18,19] and establishes an index system including hazard, vulnerability, exposure, and resilience for multi-disaster integrated risk assessment. Exposure refers to exposure units (people, property, systems, or other elements present in hazard zones) that may suffer losses as a result of disasters [20]. Resilience includes hard resilience and soft resilience [9], which in this context refers primarily to soft resilience: the ability of the disaster-bearing body to withstand, absorb, accommodate to, and restore from the effects of the disaster in a timely and efficient manner.

Different cities have different resilience to disasters [13,21,22]. At present, there are relatively few studies on the integrated risk assessment of urban land space natural disasters, man-made disasters, land resilience, and post-disaster recovery. Previous studies mainly focused on the risk assessment of a single disaster, while this study takes the integrated analysis of multiple disasters as the background, and assesses the hazard, vulnerability, exposure, and post-disaster recovery ability of the disaster-bearing body, thus, the problem that it is impossible to carry out a unified quantitative evaluation of all disaster-causing factors has been solved, and an important scientific basis has been provided for the compilation of territorial space planning. In order to improve the policy alternatives logically and test the multi-disasters assessment system, the study takes Jinan, Shandong Province as a case, based on the analytic hierarchy process (AHP), the Grey Relation Analysis (GRA), China probabilistic seismic hazard analysis (CPSHA) [23] and other methods to determine the multi-disaster integrated risk level of Jinan. The exposure, vulnerability, urban resilience, and multi-disaster integrated risk in Jinan were assessed, and suggestions for urban development and spatial planning were discussed.

## 2. Study Area

Jinan City is located in the center of Shandong Province, China. Its geographical location is between 36°02′–37°54′ N and 116°21′–117°93′ E. It is located at the intersection of the low-lying hills of central and southern Shandong and the alluvial plain of northwestern Shandong, with high-lying terrain in the south and low-lying in the north (Figure 1). Jinan consists of twelve districts or counties: Lixia, Shizhong, Huaiyin, Tianqiao, Licheng, Changqing, Zhangqiu, Jiyang, Laiwu, Gangcheng, Pingyin, and Shanghe, with a total area of 10,244.45 km^2^. By the end of 2021, Jinan had a permanent population of 6.34 million and annual GDP of about RMB 1140 billion (http://english.jinan.gov.cn/, accessed on 1 October 2022). The main urban areas of Jinan include the new urban areas and the old urban areas, mainly composed of Huaiyin, Tianqiao, Licheng, Shizhong, and Lixia [24]. The urban areas cover about 25% of Jinan’s population, 40% of its GDP, and 19% of its buildings.

Major natural disasters in Jinan include geological disasters, meteorological disasters, earthquakes, and floods [25]. The hilly areas of Jinan are prone to landslides, mudslides, and other geological hazards, and there are about 300 potential geological disaster sources in the city (Figure 2). The high incidence period of geological disasters in Jinan is from June to September each year, and 27 geological disasters were caused by Super Typhoon Lekima in 2019 alone. Meteorological Disasters in Jinan are mainly caused by droughts and cold waves, causing huge losses to agriculture and the economy. Jinan is located between two large active fault zones, Tanlu and Liaokao. In addition, there are many other small fault zones in the region, so there is a geological tectonic setting in which moderate and strong earthquakes occur. There were 40 earthquakes in Jinan alone in 2020. The characteristics of seismic activity are small in size, low in magnitude, and high in frequency. From 2010 to 2020, there were 36 torrential rains and floods in Jinan. In 2007, heavy rains in Jinan caused moat water into the city, causing 330,000 people to be affected, and seriously affecting the safety of people’s lives and health and the sustainable development of the city. Man-made disasters in Jinan include fires, road traffic accidents, and so on. At present, Jinan has four chemical industrial parks, more than 2200 hazardous chemicals, and inflammable and explosive enterprises. Therefore, Jinan is a city with complex disaster situations. The multi-disaster integrated risk assessment can help to determine which areas have high disaster risks and promote the construction of a disaster risk early warning system. At the same time, the assessment can guide the construction of a resilient city and fundamentally improve the ability to resist disasters.

## 3. Methods

In this study, data such as meteorological station data, remote sensing data, and land use/land cover (LU/LC) data were used. The data used by the assessment mainly came from the Jinan Public Data Open Network (http://data.jinan.gov.cn/jinan/index, accessed on 7 October 2022) and the Jinan Survey and Mapping Institute. Detailed descriptions of the methods or formulas used in the study are provided in each section.

### 3.1. Methods of Disasters Hazard Assessment

#### 3.1.1. Hazard Assessment Method of Geological Disaster

In this study, based on the Grey Relation Analysis (GRA), the spatial distribution of geological disaster in Jinan was assessed by using meteorological stations data, geological data (such as source points of geological disaster, the spatial distribution of goaf areas, etc.), field survey data and LU/LC. The data of collapse, landslide, debris flow, karst collapse, goaf collapse, ground fissure, and 10 m land subsidence were selected as the correlation indexes in GRA analysis, the correlation indexes were classified according to the level of disaster types, and the gray correlation coefficient and correlation degree of each correlation index was calculated based on Formulas (1) and (2).
(1)$$\xi_{i}\left(k\right)=\frac{\underset{i}{\min}\;\underset{k}{\min}|x_{0}\left(k\right)-x_{i}\left(k\right)|+\rho \cdot \underset{i}{\max}\;\underset{k}{\max}\left|x_{0}\left(k\right)-x_{i}\left(k\right)\right|}{\left|x_{0}\left(k\right)-x_{i}\left(k\right)\right|+\rho \cdot \underset{i}{\max}\;\underset{k}{\max}\left|x_{0}\left(k\right)-x_{i}\left(k\right)\right|}$$
(2)$$Cor_{i}=\frac{1}{n}\overset{n}{\underset{i=1}{\sum }}\xi_{i}\left(k\right)$$
where, $\xi_{i}\left(k\right)$ denotes the grey relation coefficient of the *i*-th index and the *k*-th rank; $x_{0}$ is the custom reference matrix, $x_{i}\left(k\right)$ represents the value of the *i*-th index and the *k*-th rank; ρ  is an adjustable coefficient, the range of values is (0, 1), generally, 0.5 is more suitable; $Cor_{i}\;$ is the correlation of the *i*-th index.

#### 3.1.2. Hazard Assessment Method of Earthquake

The hazard level of the earthquake disaster in Jinan was assessed by the China probabilistic seismic hazard analysis(CPSHA), combined with field investigations, LU/LC. CPSHA is an improved probabilistic seismic hazard analysis (PSHA) [26], which is applied to the study of the ground motion parameters of the fifth-generation earthquake in China [23], etc. The basic steps are as follows:(1)The seismicity level of the seismic belt is measured by analyzing the seismicity of the seismic belt as a statistical unit.(2)Determine the magnitude distribution. Assume that seismic activity follows the G-R relationship:
(3)$$\mathrm{lg}N\left(M\right)=a-bM$$

In the formula (3), $N\left(M\right)$ is the number of earthquakes occurring in a certain period between small areas (*M* ± Δ*M*) centered on magnitude *M*; *a* and *b* are constants, *a* represents the level of seismic activity in the statistical time and region, and the *b* value represents the proportional relationship between the number of large and small earthquakes in the region. Take a probability density function from the magnitude-frequency relationship:(4)$$f\left(M\right)=\frac{\beta \exp[-\beta \left(M-M_{0}\right)]}{1-\exp[-\beta \left(M_{u}-M_{0}\right)]}$$
where, β=bln10, *b* is the slope of the magnitude frequency relationship, *M*_0_ and *M_u_* are the lower and upper magnitude limits, respectively, and *M* is the magnitude bin.

(3)According to different structures, different potential sources are divided. In the well-divided areas, the earthquake occurrence satisfies the assumption of uniform distribution, but the occurrence may be different in different focal areas.(4)It is assumed that the number of earthquakes in the potential source area accords with Poisson distribution.(5)$$P\left(A\geq a\right)=1-\exp(-v_{0}\overset{\underset{}{}}{f\left(M\right)f(r|M)\begin{array}{c} M_{u}\\ P\\ M_{0} \end{array}(A\geq a|M,r)drdM}$$
where usually *M*_0_ = 4. Among the earthquake magnitude, the first to the third magnitude will not cause damage to buildings, because only earthquakes of magnitude 4 or above are considered in engineering.

(5)Combining the effects of all statistical areas, if *N_s_* statistical areas are related to the site, the total exceeding probability is:(6)$$P\left(A\geq a\right)=1-\overset{N_{S}}{\underset{k=1}{\prod }}(1-P_{k}\left(A\geq a\right))$$
based on all the above formulas, the basic formula of CPSHA method is obtained:(7)$$\begin{array}{l} P_{i}\left(A\geq a\right)\\ =1-\exp\{-\frac{2v_{0}}{\beta }\overset{N_{m}}{\underset{j=1}{\sum }}\overset{N_{S}}{\underset{i=1}{\sum }}P(A>a|E)*f\left(\theta \right)\frac{f_{i,mj}}{A\left(s_{i}\right)}sh\left(\frac{1}{2}\beta \Delta m\right)d_{x}d_{y}d_{\theta }\} \end{array}$$
where, $A\left(s_{i}\right)$ is the area of the *i*-th potential source area in the seismic zone, $P\left(A>a|E\right)\;$ is the probability that the ground motion at a long point exceeds a for a specific earthquake time in the *i*th potential source area in the seismic zone, and $f\left(\theta \right)$ is the probability density function of the rupture direction. Based on the CPSHA calculation results, the influence areas of different seismic activity zones, the active areas of seismic activity, and the seismic fortification standards of different districts and counties were also considered to assess the earthquake hazard level in Jinan.

#### 3.1.3. Hazard Assessment Method of Flood

The occurrence of flood and waterlogging disasters is closely related to rainfall intensity, topographic complexity, river density, and other factors [27,28]. In this study, the intensity of floods at Jinan’s meteorological stations during the rainy season (May-September) was calculated by using the degree of rainfall deviation from the average rainfall. Flood intensity, elevation, slope, and distance from rivers were used to assess the hazard of flood and waterlogging disasters in Jinan. The method of calculation is shown in Formula (8).
(8)$$Flood=T\times W_{1}+H\times W_{2}+S\times W_{3}+D\times W_{4}$$
where, *Flood* is Flood hazard, *T* is Flood intensity, *H* is elevation, *S* is slope, *D* is distance from river. *W_i_* is each index weight, which were 0.3, 0.2, 0.2, and 0.3, respectively [29,30].

#### 3.1.4. Hazard Assessment Method of Meteorology Disaster

Drought and heavy rain are the most frequent meteorological disasters in North China, which have a great impact on agriculture and the economy [31]. Jinan is also known as the “Spring City”, both drought and heavy rain have an impact on Jinan’s spring water level and tourism revenue. In this study, the Normalized Vegetation Supply Water Index (*NVSWI*) (Formula (9)) was used to assess the hazard of drought in Jinan. The correlation between *NVSWI* and soil moisture is high, and the evaluation effect of drought is good [32].
(9)$$VSWI=\frac{NDVI}{LST}$$
(10)$$NVSWI=\frac{VSWI-VSWI_{min}}{VSWI_{max}-VSWI_{min}}$$
where, land surface temperature (*LST*) was extracted from the MOD11A2 data product, normalized difference vegetation index (*NDVI*) was extracted from the MOD11A3 data product. $VSWI_{min}$ and $VSWI_{max}$ represent the minimum and maximum pixel values of *VSWI* at a given time period, respectively.

Based on the rainfall data of Jinan, the rainfall intensity >30 mm per 12 h or >50 mm per 24 h is defined as the rainstorm standard to calculate the annual mean rainfall and frequency of heavy rainfall in Jinan. Based on Formulas (11) and (12), the hazard of heavy rainfall in Jinan was assessed:(11)$$D=R_{td}\times R_{n}$$
(12)$$R_{td}=\frac{R_{t}-R_{tmin}}{R_{tmax}-R_{tmin}}\times \left(R_{nmax}-R_{nmin}\right)+R_{nmin}$$
where, *D* is the hazard index of the rainstorm, *R_td_* is the standardized annual rainstorm, *R_n_* is the frequency of rainstorm, $R_{t}$, $R_{tmax}\;$, and $R_{tmin}$ are the actual, maximum, and minimum of the annual rainstorm, respectively, $R_{nmax}$, and $R_{nmin}$ are the maximum and minimum of the annual mean frequency of rainstorm. Finally, the hazard level of meteorological disasters in Jinan was obtained by taking into account the heavy rainfall and drought conditions in Jinan.

#### 3.1.5. Risk Assessment Method of Fire

Based on the data of Jinan’s fire sources (petrochemical enterprises, high-pressure pipelines, key forest protection areas, etc.), the fire hazard level was assessed by Formula (13).
(13)$$F=C\times W_{1}+T\times W_{2}$$
where, *F* denotes the hazard of fires, *C* denotes the hazard of city fire, and *T* denotes the hazard of a forest fire; $W_{1}\;$ and $W_{2}$ represent weight, which are 0.6 and 0.4, respectively [29,33].

### 3.2. Index System

The Analytic Hierarchy Process (AHP) has been widely used in disaster integrated risk assessment since the 1970s [7,34,35,36]. In this study, based on AHP and multi-disaster integrated risk assessment index system, selecting earthquake, geological disaster, meteorological disaster, floods, and fire as the Index of risk factors, the multi-disaster Integrated Risk was calculated by “Hazard*Exposure*Vulnerability/Resilience”. Exposure is related to geographical entities (population, built-up areas, infrastructure, etc.) of the area [17,37], therefore, population density, building density, economic density, and infrastructure density were selected as indicators of disaster exposure in Jinan. Combined with other studies [18,38] and the actual situation in Jinan, it can be seen that the elderly, children, and agriculture are more vulnerable to natural or man-made disasters. Therefore, the proportion of the old and young population and the proportion of the primary industry were selected as vulnerability assessment indicators. The main indexes of urban resilience include emergency evacuation ability, emergency water supply ability, emergency medical facilities, emergency fire fighting facilities, emergency placement ability, and density of village (community). The study used ArcGIS software to assign vector data of different levels of medical stations, fire stations, and river networks, and get the data of each index layer of urban resilience after rasterization through spatial kernel density analysis, Euclidean distance, and other spatial analyst tools.

Delphi method is an anonymous, feedback and statistical expert survey method, which ultimately obtains a unified opinion through continuous anonymous feedback of expert suggestions. AHP makes a quantitative analysis of qualitative indicators by constructing a multi-level evaluation index system. The combination of the Delphi method and AHP can quickly establish the evaluation index system, avoid the interference of experts’ opinions, and realize the calculation of indicators and weights. Based on the combination of the Delphi Method [39] and AHP, 17 quantitative indexes were identified, and the weight of each index was determined according to the level of contribution of each index to its category layer, as shown in Table 1.

## 4. Results

### 4.1. Result of Disaster Hazard Assessment

Based on Formulas (1)–(13), combined with LU/LC and field survey data, the hazard level of each type of disaster was assessed and the results were shown in Figure 3a–e, where the value range of low, medium and high hazard levels are 0–20, 20–30 and 30–40 respectively. The disaster hazard assessment results showed that the central region of Jinan had the higher hazard level of geological disaster, meteorological disaster, and fire, while the south-eastern region of Jinan had the higher hazard level of all five disasters, the northern and southwestern parts of Jinan are at low hazard for all five disasters. Jinan’s main urban area is characterized by large topographical changes and complex geological structures. As the city expands, traffic networks and urban housing are built, and mountains are mined and excavated, collapse, landslide, and debris flow disasters occur frequently in the main urban area. The area with high and medium geological hazard risks is about 410 km^2^, accounting for 60% of the total urban area. Zhangqiu is a coal-producing area, that once had more than 200 coal mining enterprises of various sizes. Decades of coal mining have made Zhangqiu prone to goaf collapse or geological collapse. Gangcheng used to focus on the development of the steel and coal industry, and there was a situation of coal mine goaf collapse. Therefore, the hazard level of geological disasters in the middle and southeast of Jinan is relatively high. The regional distribution of high seismic hazard levels is closely related to the distribution of seismic fault zones and seismically active points. The seismically active points in Jinan are relatively concentrated in the southeast and a small area in the west of Changqing. According to the data from meteorological stations, the flood intensity in Gangcheng is higher than in other counties in recent years, so the hazard level of flood in the south of Gangcheng is high. The meteorological disasters in Jinan are mainly drought, and the low-value areas of NVSWI are mainly located in the Jinan urban area and Laiwu. The vegetation coverage in the urban area is low, and the LST is high because of the urban heat island. Especially in summer, the surface reflection causes urban high temperatures, and the surface water level and groundwater level decrease [40]. The distribution density of logistics warehousing enterprises and flammable and explosive points in the Jinan urban area, Zhangqiu and Laiwu are high, and some flammable and explosive points are located in areas with high forest vegetation coverage, so the fire hazard level in Jinan presents the distribution pattern shown in Figure 3e.

In this study, through the comprehensive analysis of the hazard level of five disasters, the disaster hazard assessment results of Jinan City are obtained in Figure 3f. On the whole, most areas of Jinan City are at a low level, and medium and high levels are mainly located in Zhangqiu, Laiwu, Gangcheng, Huaiyin, Shizhong, Lixia, and Licheng, accounting for about 17% of the total area of Jinan. This part of the region has a high probability of disasters, and its territory spatial planning should focus on emergency disaster reduction.

### 4.2. Result of Exposure Assessment

The higher the density of the disaster-bearing body, the higher the regional exposure. It can be seen from Figure 4 that the exposure levels of different regions in Jinan vary greatly, where the value range of low, medium, and high exposure levels are 0–20, 20–30, and 30–40 respectively. Generally, low-level exposure is the main exposure. The areas with medium and high levels of exposure account for about 20% of the total area of Jinan, mainly concentrated in the urban areas of Jinan and the urban areas of various districts and counties. The population density, economic density, building density, and infrastructure density of these areas are higher, and the exposure to disasters is more complex. Compared with Figure 3, it can be found that although the area proportion of areas with high and medium hazard levels in Jinan is small, the exposure level of that is high, and the population and buildings are concentrated in areas with high and medium hazard levels. Based on the analysis of ArcGIS, 1.4% of the population, 1.1% of GDP, and 8.95% of buildings in Jinan are exposed to the high-risk level, with 28.0% of the population, 34.6% of GDP, and 65.8% of buildings are exposed to the medium hazard level. About 10.2% of Jinan is low-level regions of hazard, but disaster exposure is medium or high level. Although the hazard level of this region is low, it is easy to cause huge casualties and economic losses in case of disasters, so the territory spatial planning for this region should also focus on preventing natural disasters and man-made disasters.

### 4.3. Result of Vulnerability Assessment

The vulnerability assessment results of Jinan City are shown in Figure 5, where the range of values of low, medium, and high vulnerability levels are 0–20, 20–30, and 30–40, respectively. The overall vulnerability of disaster-bearing bodies in Jinan is low level, and the low vulnerability level areas account for about 71% of the total area of Jinan. The areas with the vulnerability of medium level are mainly distributed in other districts and counties except for the urban area of Jinan. The proportion of primary industry in these areas is high. Food crops and characteristic cash crops (The Zhangqiu Green Onion, etc.) are vulnerable to drought, flood, hot dry wind, and cold waves. The proportion of the elderly and young people in the urban area of Jinan is high, but the proportion of the primary industry is small, so the urban area is not very susceptible to the destructive impact of disasters. The areas with a high level of vulnerability only account for 0.5% of the total area of Jinan, scattered mainly in Zhangqiu District and Licheng District. Among the regions with medium and high vulnerability levels, 18.6% of the regions also have medium and high hazard and exposure levels, which are scattered throughout the city. This characteristic of scattered distribution in space will challenge disaster reduction and territory spatial planning in the future.

### 4.4. Result of Resilience Assessment

The overall resilience of Jinan is at an optimistic level (Figure 6), where the range of values of low, medium, and high resilience levels are 0–20, 20–30, and 30–40, respectively. The area with a high level of resilience accounts for about 20% of the whole area of Jinan, which is mainly distributed in the urban areas of Jinan and the middle of all districts and counties. The areas of medium-level resilience ring-shaped around the areas with high-level resilience. This spatial pattern is closely related to the distribution of the road network, water system, medical infrastructure, and temporary placement points in Jinan. The area with a low level of resilience accounts for 7.79% of the total area of Jinan, mainly distributed in the areas bordering Jinan and other cities, or mountains. The population and infrastructure in these areas are less distributed, so resilience is weak. Although the resilience level in the southwest of Changqing and the north of Laiwu is low, the hazard level, disaster vulnerability level, and disaster exposure level in these two regions are low, so the possibility of disasters and losses is small. Therefore, it is unnecessary to invest too much money and manpower in disaster prevention and mitigation in these two regions.

### 4.5. Result of Multi-Disaster Integrated Risk Assessment

On the basis of Section 4.1, Section 4.2, Section 4.3 and Section 4.4, the multi-disaster integrated risk level of Jinan was assessed according to the index weights determined by AHP, and the results are shown in Figure 7, where the range of values of low, medium, and high multi-disaster integrated risk levels are 0–20, 20–30, and 30–40, respectively. The multi-disaster integrated risk of Jinan is generally at a low level. The areas with low integrated risk levels account for 85.69% of the total area of Jinan, while the areas with medium and high integrated risk levels account for 14.31% of the total area. Medium integrated risk regions are mainly distributed in the south of Licheng, the west of Changqing, Zhangqiu, etc. High-integrated risk regions are mainly distributed in Zhangqiu, totaling 43.57 km^2^, and scattered in Changqing, Laiwu, and Licheng. Based on ArcGIS analysis, about 24.9% of Jinan’s GDP, 21.1% of its population, and 41.8% of its buildings are located in medium and high-integrated-risk-level regions. The area with a medium and high level of integrated risk in the urban area accounts for 34.5% of the total urban area. In the main urban area, 39.2% of the population, 39.3% of the GDP, and 49.1% of the buildings are at the medium and high multi-disaster integrated risk level. Although the multi-disaster integrated risk level in the north and south of the urban area is low, the Yellow River in the north and the mountains in the south hinder the urban expansion, which has caused great trouble to the territory’s spatial planning and urban development.

## 5. Discussion

The multi-disaster integrated risk assessment of Jinan can analyze the disaster hazard level of geological disasters, floods, meteorology disasters, earthquakes, and fire, and to a certain extent, define the exposure of disaster bearing bodies, the vulnerability of disaster bearing bodies, the resilience of the city and the risk of multi-disaster, which also helps to provide suggestions and guidance for urban emergency disaster reduction and territorial spatial planning. Based on the assessment results (Figure 3, Figure 4, Figure 5, Figure 6 and Figure 7), we further put forward relevant suggestions on disaster prevention and territorial spatial planning of Jinan and discussed the current shortcomings and further research in the future.

### 5.1. Suggestions and Measures for Disaster Prevention and Mitigation

In the context of global climate change, extreme weather, and climate events such as high temperatures and heavy rainfall in Jinan will increase and intensify, and the hazard level of suffering from drought, flood, and other natural disasters will increase, causing more economic losses [41]. At the same time, with the acceleration of the urbanization process of Jinan, the population inflow into Jinan from surrounding cities, the population, economy and building density, and other disaster bearing bodies will further increase, and the exposure level in Jinan will also show an increasing trend. However, population mobility and economic development will also reduce the proportion of the elderly and young population and the proportion of the primary industry in GDP to a certain extent, reducing the vulnerability level of Jinan. Therefore, in the future, Jinan should focus on improving the level of resilience, while reducing the level of hazard, exposure, and vulnerability. According to the current assessment results, the key areas for disaster prevention and reduction in Jinan City are urban areas, Zhangqiu, Laiwu, and Gangcheng. The main disasters to be prevented should be geological disasters, meteorological disasters, and fire. The specific prevention and control measures should include:Based on the results of the geological hazard risk analysis in this study, the Jinan government should allocate special funds to carry out engineering measures such as cutting slopes and building walls at the geological collapse sites. The results show that the hazard level of geological disasters in the urban area of Jinan, Laiwu, and Zhangqiu is relatively high, so it is necessary to carry out a geological survey for the mine goafs in Zhangqiu and Laiwu, clarify the distribution range of the goafs, predict the ecological environment and geological problems of the existing mines, hold expert seminars to propose reasonable and efficient mine restoration and management plans.The research result shows the hazard level of an earthquake in Jinan is low, but the earthquake disaster has a wide range of impacts, is easy to break out, and is difficult to predict. Therefore, the prevention of earthquake disasters should focus on the seismic evaluation and reinforcement of buildings, comprehensively checking the seismic grade of urban houses, schools, reservoirs, dams, and dangerous chemical plants in Jinan, and reinforcing or rebuilding old houses that cannot meet the seismic fortification requirements in urban and rural areas. At the same time, the results show the level of an earthquake in the southeast is relatively high, so the layout of seismic stations should be optimized, and active fault detection should be carried out.The hazard level of floods is low, and the occurrence of urban flood disasters can be effectively prevented by carrying out protection and control work on the Yellow River, Xiaoqing River, Baiyun Lake, etc. Combining big data intelligent analysis and 5G technology, improve the intelligence of the hydrological monitoring network and do a good job in flood disaster prediction and early warning in Laiwu.The government should focus on strengthening the construction of irrigation, drainage, power, and transportation lines in agricultural planting areas with high meteorological disaster hazards levels, such as the south of Laiwu and the middle of Zhangqiu, to mitigate the impact of drought, rainstorm, and cold wave on agricultural production.The study shows that the hazard of fire in Jinan presents a spatial distribution trend of spreading from the urban area to the surrounding areas. The government should strengthen the supervision of flammable and explosive materials in urban areas and districts, counties, and cooperate with grass-roots mass autonomous organizations to avoid fire.

### 5.2. Suggestions and Measures for Territorial Spatial Planning

Disasters are hindering the development of cities, but at the same time, they are also urging the reform and innovation of territorial spatial planning. The result of the multi-disaster integrated risk assessment provides new suggestions and requirements for territorial spatial planning. The government should pay attention to the importance of the integrated risk assessment for territorial spatial planning, and formulate more targeted planning to reduce planning costs and improve the scientificity of planning. Based on the research results, it can be seen that by delineating the spatial distribution of disaster hazard levels and integrated risks level in Jinan, the planning for the medium and high integrated risk level areas that only account for 14.3% of Jinan’s total area, can serve 24.9% of the city’s population, 21.1% of the city’s GDP and 41.8% of the city’s buildings. In addition, the government should make full use of the research results to monitor the high-risk level areas of various disasters within the areas of permanent basic farmland, ecological protection red line, and urban development boundary of Jinan, so as to adjust the spatial distribution of them in the next planning period.

### 5.3. Research Deficiencies and Prospects

There are still some deficiencies in the current research. In the disaster risk assessment, based on the actual situation of Jinan and other experts’ research [5,18,27,37], five major disasters were identified, without considering hot, dry wind, hail, haze, and other disasters. Taking hot, dry wind disasters as an example and referring to the research of others [42], Zhangqiu, Licheng, Laiwu, Gangcheng, Jiyang, and Shanghe are located in the high dry, hot wind disaster area. As an agricultural wind disaster, dry and hot wind will lead to the evaporation of humidity in the air and soil moisture, intensify plant transpiration, and seriously affect crop yield and economic development [42,43]. Therefore, further analysis of other disasters is needed in the following research to improve the comprehensiveness and richness of comprehensive disaster assessment. In addition, the multi-disaster integrated risk assessment system has a clear structure and explicit method, but when it is applied to different cities, there are differences in data collection channels, geographical location, and climate conditions. Therefore, some indicators or weight values should be modified in the application of different cities to reduce the error generated and improve the scalability of the method when the method is applied to other cities.

## 6. Conclusions

In this study, we identified a multi-disaster integrated risk assessment system. Firstly, based on the GRA, CPSHA, and other methods, the hazard of geological disasters, meteorological disasters, flood disasters, earthquakes, and fires were evaluated. Then, based on the AHP method, the level of hazard, exposure, vulnerability, resilience, and multi-disaster integrated risk were assessed. Finally, the relevant suggestions on disaster prevention and reduction and territorial spatial planning were put forward. Taking Jinan as an example, the multi-disaster integrated risk assessment system is analyzed and explained, and we draw the following conclusions:The hazard level of geological disasters, meteorological disasters, and fires in the central part of Jinan is relatively high. The risk level of geological disasters, earthquakes, floods, meteorology, and fires in the southeast part of Jinan is high, while the risk level of disasters in other areas is low.Jinan has a low level of disaster hazard, exposure, and vulnerability, and a high level of resilience. The areas with high disaster exposure are concentrated in central Jinan and Laiwu, the areas with a high level of disaster vulnerability are scattered throughout the city, and the areas with high resilience levels are mainly concentrated in urban areas of Jinan and urban areas of all districts and counties. The overall level of multi-disaster integrated risk in Jinan is at a medium or low level, the high-level areas are mainly distributed in the urban areas of Jinan, Zhangqiu, and Laiwu.24.9% of Jinan’s GDP, 21.1% of its population, and 41.8% of the buildings are distributed in the medium and high multi-disaster integrated risk level areas that account for 14.3% of Jinan’s total area. The risk assessment results of Jinan are of great significance to the implementation of disaster prevention and mitigation projects and the compilation of territorial spatial planning, which can effectively reduce the planning cost and improve scientific and effective planning.

## Figures and Tables

**Figure 1 ijerph-20-03483-f001:**
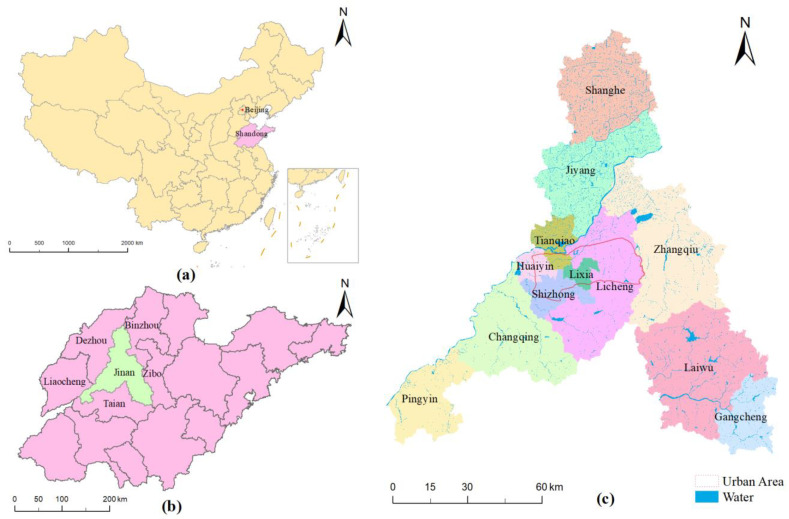
Geographical location map of the study area:(**a**) location of Shandong Province in China; (**b**) location of Jinan City in Shandong Province; (**c**) location of districts and counties in Jinan.

**Figure 2 ijerph-20-03483-f002:**
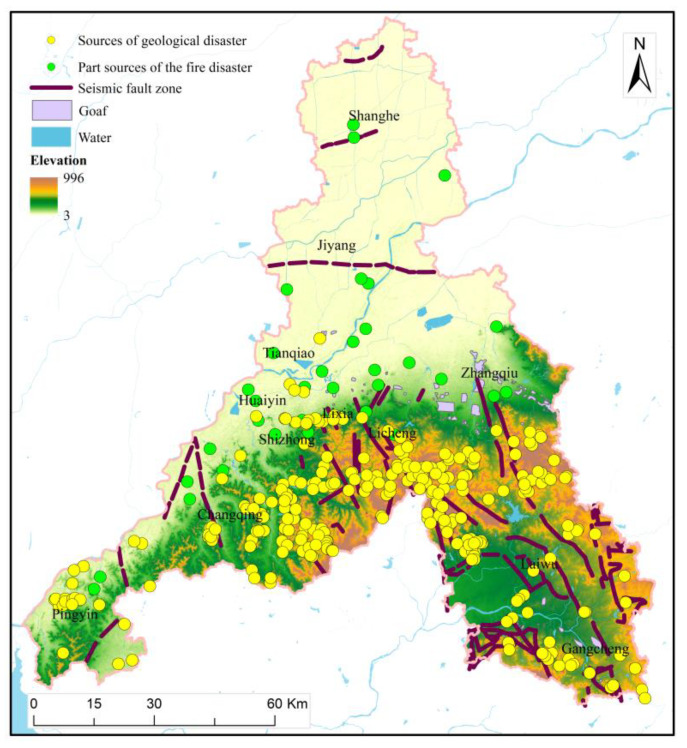
Part of disaster sources in Jinan.

**Figure 3 ijerph-20-03483-f003:**
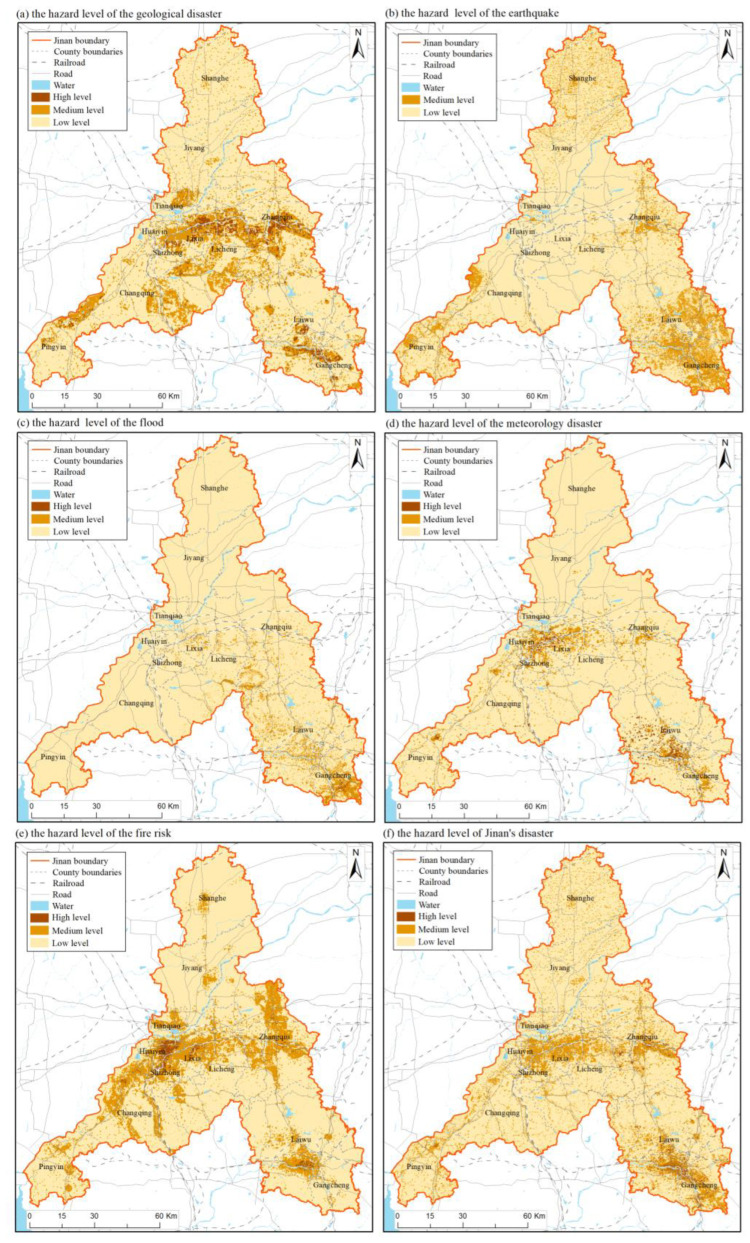
Result map of hazard assessment.

**Figure 4 ijerph-20-03483-f004:**
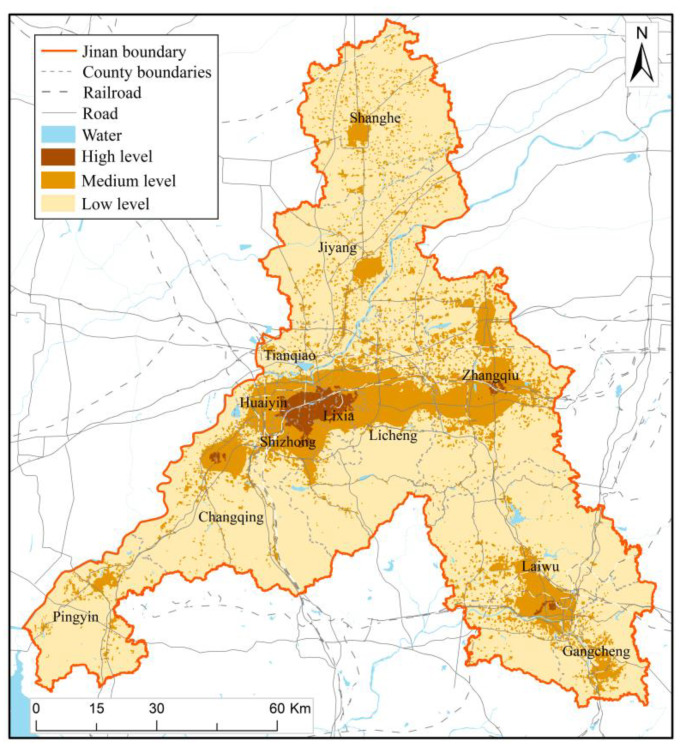
Result map of exposure assessment.

**Figure 5 ijerph-20-03483-f005:**
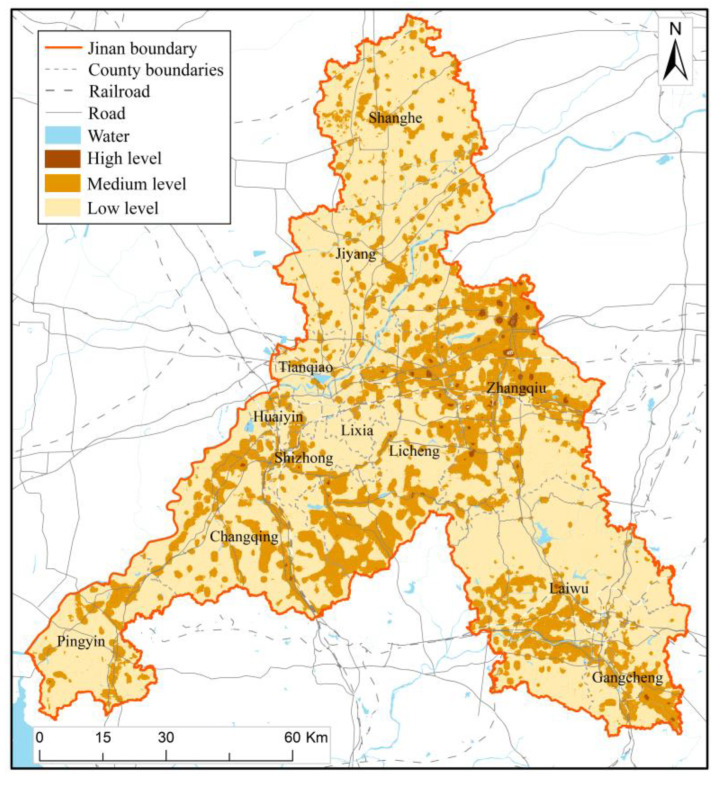
Result map of vulnerability assessment.

**Figure 6 ijerph-20-03483-f006:**
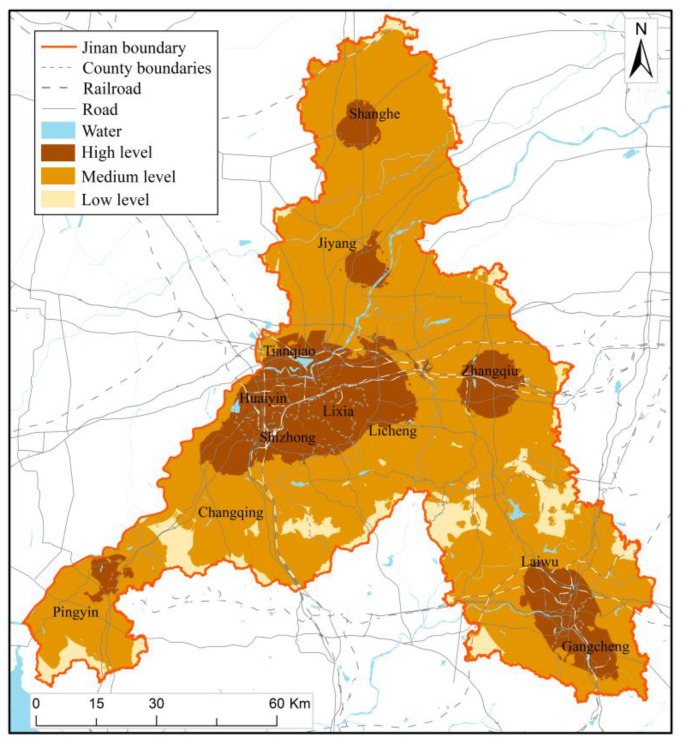
Result map of resilience assessment.

**Figure 7 ijerph-20-03483-f007:**
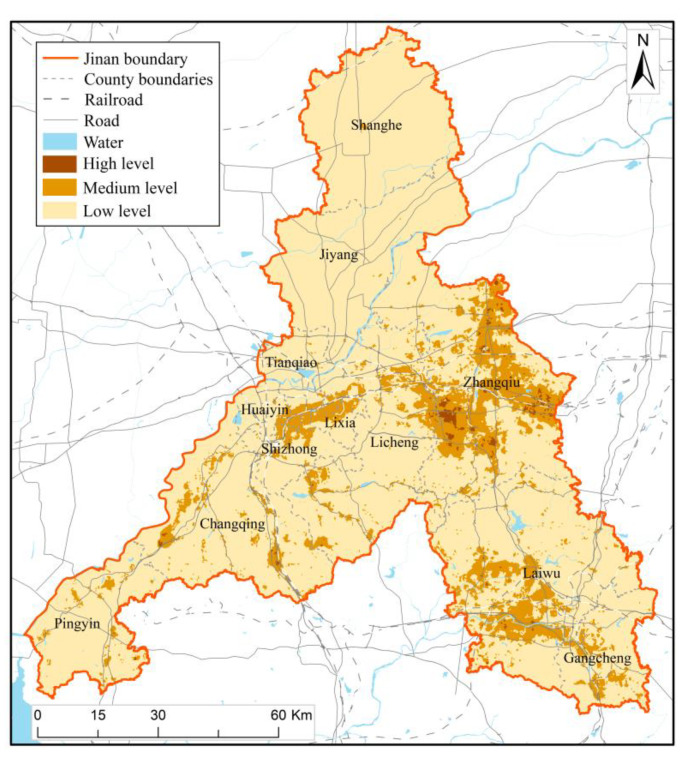
Result map of multi-disaster integrated risk assessment.

**Table 1 ijerph-20-03483-t001:** Index system of multi-disaster integrated risk assessment.

Category	Weight	Index	Weight
Hazard	0.0738	Risk of geological disaster	0.0172
		Risk of earthquake	0.0161
		Risk of floods	0.0145
		Risk of meteorological disaster	0.0131
		Risk of fire risk	0.0129
Exposure	0.1482	Population density	0.0429
		Economic density	0.0303
		Building density	0.0607
		Infrastructure density	0.0143
Vulnerability	0.1889	Proportion of primary sector of the economy	0.1567
		Proportion of old people and children	0.0322
Resilience	0.5891	Emergency evacuation capability ^1^	0.1281
		Emergency water supply capability ^2^	0.0848
		Emergency fire fighting facilities	0.0383
		Emergency medical facilities	0.2095
		Emergency placement capacity	0.0643
		Density of village (community)	0.0641

^1^ Based on ArcGIS Kernel Density, emergency evacuation capability was calculated using Jinan’s road network data. ^2^ Based on ArcGIS Euclidean distance analysis, emergency water supply capability was calculated using Jinan’s river system data.

## Data Availability

Not applicable.

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
