# Peer review of "Multi-Disaster Integrated Risk Assessment in City Range—A Case Study of Jinan, China"

_ijerph, 2023, doi:10.3390/ijerph20043483_

Round 1

Reviewer 1 Report

This manuscript proposes a multi-disaster integrated risk assessment system for cities. This is an important issue, because, as authors say, 70-80% of gross domestic product (GDP) and population of the region is concentrated in cities. Outcomes of this kind of studies are useful for stakeholders involved in decision-making. However, on the one hand this manuscript have important lacks of scientific rigor. On the other hand, it focus directly in the application to the Jinan City, and does not give a clear explanation of the methodology used. In my opinion this study is poorly explained, incorrectly at some expressions and incomplete.

-According to the Varnes and IAEG Commission on Landslides (1984), risk is defined as R (I) = H × V (I) × E, where R is the landslide risk, H is the landslide hazard, V is the vulnerability of vulnerable elements, I is the intensity of landslides and E is the value of the element at risk (e.g. the number of people or the monetary value of the buildings). We can find another example for earthquakes risk assessment, where seismic hazard is the probability that an earthquake with a ground motion intensity will occur in a specific area and time.

However, the equation used by authors in the abstract and manuscript seems wrong, because multi-disaster integrated risk is evaluated from risk instead of hazard, and same in line 67 (page2, among others). Moreover, in line 48 (page2) authors define risk from vulnerability and hazard, but use a wrong definition of hazard “..damage or economic loss cause by natural or man-made disasters …”. This definition refers to risk, not hazard!  I do not understand how authors evaluate the risk without hazard.

-The vulnerability analyzed in this study is only a geographical o social vulnerability, because authors consider only the primary sector of the economy and the proportion of old people and children. However I do not understand why you consider as elements exposed the buildings, but not their vulnerability.

-Some variables of equations of section 3.1 are not correctly indicated, and not clearly explained and no units are indicated, or some variable are not explain (e.g. Rth in eq. 11-12).

-Authors give in section 5.1 some countermeasures for disaster prevention and suggestions for territorial spatial planning, which rather than being based on results, seem to be logical proposals based on knowledge of the territory, or at least, its relation to the results of the study is not clearly shown.

Based on the previous exposed issues, I consider the manuscript with title "Multi-Disaster integrated risk assessment in city range - a case study of Jinan, China " should be rejected to be publish in the International Journal of Environmental Research and Public Health.

Author Response

Point 1: According to the Varnes and IAEG Commission on Landslides (1984), risk is defined as R (I) = H × V (I) × E, where R is the landslide risk, H is the landslide hazard, V is the vulnerability of vulnerable elements, I is the intensity of landslides and E is the value of the element at risk (e.g. the number of people or the monetary value of the buildings). We can find another example for earthquakes risk assessment, where seismic hazard is the probability that an earthquake with a ground motion intensity will occur in a specific area and time.

However, the equation used by authors in the abstract and manuscript seems wrong, because multi-disaster integrated risk is evaluated from risk instead of hazard, and same in line 67 (page2, among others). Moreover, in line 48 (page2) authors define risk from vulnerability and hazard, but use a wrong definition of hazard “..damage or economic loss cause by natural or man-made disasters …”. This definition refers to risk, not hazard!  I do not understand how authors evaluate the risk without hazard.

Thank you very much for your comments. According to your comments, we found that there are certain deficiencies in the research. We have made relevant explanations and amendments according to your suggestions. First of all, you pointed out that the multi-disaster comprehensive risk analysis is based on the hazard analysis, and in our subsequent analysis, it is based on the hazard analysis. This is because we have caused a lot of objections in the translation process. There are errors in the expression process, and we have corrected them according to your suggestions. Then regarding the definition of hazard in the original line 48 (page 2), the definition of formula 1 in the paper referred to here is as follows(Comprehensive flood risk assessment based on set pair analysis-variable fuzzy sets model and fuzzy AHP | SpringerLink): hazard is the function of scale and probability of natural disaster to cause harm or loss in a specified location in a period of time, thus it reflects the natural properties of the disaster. In the subsequent analysis process, the hazard was evaluated, but there were some problems in the language expression, we revised it according to your suggestion.

Point 2: The vulnerability analyzed in this study is only a geographical o social vulnerability, because authors consider only the primary sector of the economy and the proportion of old people and children. However I do not understand why you consider as elements exposed the buildings, but not their vulnerability.

Thank you very much for your comments. At the beginning of the study, we referred to the studies carried out by other scholars in China. Some studies selected population density, per capita GDP, construction age, etc. as the indicators of vulnerability analysis. We believe that vulnerability research is to analyze the degree of vulnerability of social and geographical environment. China is a developing country, especially the relatively low level of development in northern China, Often areas with high population density and high economic density are areas where young people and high-tech industries are concentrated. These areas are more vulnerable to disaster images than agricultural areas and areas where non-working people are concentrated. Therefore, the proportion of the elderly and young people and the proportion of the primary industry are selected as indicators for vulnerability analysis.

Point 3: Some variables of equations of section 3.1 are not correctly indicated, and not clearly explained and no units are indicated, or some variable are not explain (e.g. Rth in eq. 11-12).

Thank you very much for your suggestion. According to your suggestion, we have corrected the missing variable description and formula in 3.1

Point 4: Authors give in section 5.1 some countermeasures for disaster prevention and suggestions for territorial spatial planning, which rather than being based on results, seem to be logical proposals based on knowledge of the territory, or at least, its relation to the results of the study is not clearly shown.

Thank you very much for your comments. According to your suggestions, we have revised the suggestions in 5.1.

Reviewer 2 Report

In considering the purpose of this research, it seems that the selection of research methodology and scope of analysis, the process of deriving alternatives, and the specificity are sufficiently systematic as an academic article. However, in order to explain the policy alternatives logically, separate drawing and presentation of implications regarding case analyses seem to be more useful. This aspect should be reflected in the modification of this article.

Author Response

Point:In considering the purpose of this research, it seems that the selection of research methodology and scope of analysis, the process of deriving alternatives, and the specificity are sufficiently systematic as an academic article. However, in order to explain the policy alternatives logically, separate drawing and presentation of implications regarding case analyses seem to be more useful. This aspect should be reflected in the modification of this article.

Thank you very much for your comments. According to your comments, we have further explained the case analysis, clarified that the case analysis is for the purpose of more comprehensive and detailed interpretation of the multi-disaster comprehensive evaluation system and the interpretation of policy options, provided practical suggestions for Jinan City, and tested the existing system.

Reviewer 3 Report

This paper proposed a multi-integrated risk assessment that considers multiple hazards and other properties such as exposure, vulnerability, and resilience. The proposed approach seems straightforward, and the results may benefit future practitioners. The information presented is organized well. However, some sections can benefit from more detailed explanations and references. Thus, minor revision is suggested before this paper can be accepted.

1.       Abstract. The equation does not need to be presented in the abstract. Instead, the findings and broader impacts of the proposed approach or the benefits of the study should be highlighted instead.

2.       Introduction. The current approach for multi-disaster risk assessment in any application should be added in the Introduction. Highlight the disadvantages of the current approaches and describe how the proposed approach can overcome the disadvantage of the current approach.

3.       The study area section is good. However, a bar chart may be presented to highlight the casualties of geological disasters. At the end of this section, identify why multi-disaster integrated risk assessment can be beneficial in reducing the casualties caused by various disasters.

4.       Methods.

-In Section 3.1.2. Line 142-155. It is better to identify the sequence of the steps instead of just using bullet points. Check formula 5 of the Mo Mu subscripts. Check formulas 5 and 6 for the bracket type. Line 157 CPHSA or CPSHA check for consistency. Additionally, all variables from formulas 3-7 should be explained. Provide a reference for M0=4, and address why M0 less than 4 is not considered.

-In Section 3.1.3. Provide references for the weight index value.

-In Section 3.1.5. Provide references for the weight index value.

-Table 1 listed the exposure, vulnerability, and resilience in the category column but the equations to quantify these three properties are not detailed enough in any of Section 3. Please provide a subsection to detail how resilience is calculated. Also, briefly introduce Delphi Method and AHP in Line 213.

5.       Results. The results for four assessments, risk, exposure, vulnerability, and resilience, are presented qualitatively with a 3-level color code (low, medium, high). However, based on the previous method sections, the assessment proposed seems to be quantitative. What do the range of values for each color-coded category for each assessment?

Author Response

  1. Abstract. The equation does not need to be presented in the abstract. Instead, the findings and broader impacts of the proposed approach or the benefits of the study should be highlighted instead.

Response1: Thank you very much for your comment. We have deleted the formula according to your suggestion and added or modified some contents.

  1. Introduction. The current approach for multi-disaster risk assessment in any application should be added in the Introduction. Highlight the disadvantages of the current approaches and describe how the proposed approach can overcome the disadvantage of the current approach.

Thank you very much for your suggestion. According to your suggestion, we mainly revised the third paragraph of the introduction, added the application cases of multi-disaster comprehensive risk assessment, and described the shortcomings of these cases, and made up for the existing deficiencies based on the concept of resilience city.

  1. The study area section is good. However, a bar chart may be presented to highlight the casualties of geological disasters. At the end of this section, identify why multi-disaster integrated risk assessment can be beneficial in reducing the casualties caused by various disasters.

Thank you again for your suggestion. In the early stage, we considered using bar chart to express the casualties of geological and fire disasters, but the bar chart could not express the spatial distribution of fire hazard points and geological disaster prone areas, so we chose to use map to express. At the end of this section, we added relevant explanations according to your suggestions

  1. Methods.

-In Section 3.1.2. Line 142-155. It is better to identify the sequence of the steps instead of just using bullet points. Check formula 5 of the Mo Mu subscripts. Check formulas 5 and 6 for the bracket type. Line 157 CPHSA or CPSHA check for consistency. Additionally, all variables from formulas 3-7 should be explained. Provide a reference for M0=4, and address why M0 less than 4 is not considered.

-In Section 3.1.3. Provide references for the weight index value.

-In Section 3.1.5. Provide references for the weight index value.

-Table 1 listed the exposure, vulnerability, and resilience in the category column but the equations to quantify these three properties are not detailed enough in any of Section 3. Please provide a subsection to detail how resilience is calculated. Also, briefly introduce Delphi Method and AHP in Line 213.

Thank you very much for your suggestion. According to your suggestion, we have revised the formula and content of the method part. According to your suggestion, we have explained the variables in the formula; For the calculation of disaster exposure, vulnerability and resilience, the spatial analysis of the obtained data, such as Euclidean distance analysis, is carried out using the ArcGIS software, and the data of each indicator layer is obtained.

  1. Results. The results for four assessments, risk, exposure, vulnerability, and resilience, are presented qualitatively with a 3-level color code (low, medium, high). However, based on the previous method sections, the assessment proposed seems to be quantitative. What do the range of values for each color-coded category for each assessment?

Thank you very much for your suggestion. We have supplemented the range of values according to your suggestion. During the research, we re-classified the experimental index layer and results, set the threshold value of the evaluation result to 0 and 40, and divided 0-20, 20-30, 30-40 into three categories: low, medium, and high.

Round 2

Reviewer 1 Report

This manuscript has been improved from its previous version where some important concepts were incorrectly used and can be published in the Special Issue "Land Management for Territorial Spatial Planning".

However, please consider sending the authors these suggestions in order to improve the manuscript:

-Given that the manuscript is completely focused on the application to the Jinan City, I consider a reference/comment to the possibility to apply this methodology to whatever city, should be included in the manuscript. In the same way.

-I think that the novelty of this study should be highlighted more in the introduction of the manuscript, compared to other previous studies.

Author Response

Point1 -Given that the manuscript is completely focused on the application to the Jinan City, I consider a reference/comment to the possibility to apply this methodology to whatever city, should be included in the manuscript. In the same way.

Thank you very much for your comment. According to your comment, we have made relevant amendments to the manuscript, which are mainly reflected in section 5.3 of the discussion section. When the method is applied to other cities, because the actual conditions of different cities are different, it is necessary to modify the relevant indicators or weights to improve the scalability of the method.

Point2 -I think that the novelty of this study should be highlighted more in the introduction of the manuscript, compared to other previous studies.

Thank you very much for your comment. We have revised the fourth paragraph of the introduction according to your comments. Compared with other studies, the main innovation of this study focuses on solving the problem that it is impossible to carry out unified quantitative evaluation of multiple disaster factors. It not only discusses the risk and vulnerability of cities when facing multiple disasters, but also discusses the exposure of disaster-bearing bodies and the resilience of cities when facing multiple disasters
